# Uptake of *Tropheryma whipplei* by Intestinal Epithelia

**DOI:** 10.3390/ijms24076197

**Published:** 2023-03-24

**Authors:** Julian Friebel, Katina Schinnerling, Kathleen Weigt, Claudia Heldt, Anja Fromm, Christian Bojarski, Britta Siegmund, Hans-Jörg Epple, Judith Kikhney, Annette Moter, Thomas Schneider, Jörg D. Schulzke, Verena Moos, Michael Schumann

**Affiliations:** 1Department of Cardiology, Angiology and Intensive Care Medicine, Deutsches Herzzentrum der Charité, 12203 Berlin, Germany; 2Department of Gastroenterology, Infectiology and Rheumatology, Campus Benjamin Franklin, Charité—Universitätsmedizin Berlin, Corporate Member of Freie Universität Berlin and Humboldt-Universität zu Berlin, 12203 Berlin, Germany; 3Berlin Institute of Health at Charité—Universitätsmedizin Berlin, 10117 Berlin, Germany; 4Departamento de Ciencias Biológicas, Facultad de Ciencias de la Vida, Universidad Andrés Bello, Santiago 8370146, Chile; 5Institute of Clinical Physiology, Campus Benjamin Franklin, Charité—Universitätsmedizin Berlin, Corporate Member of Freie Universität Berlin and Humboldt-Universität zu Berlin, 12203 Berlin, Germany; 6Institute for Microbiology, Infectious Diseases, and Immunology, Biofilmcenter, Campus Benjamin Franklin, Charité—Universitätsmedizin Berlin, Corporate Member of Freie Universität Berlin and Humboldt-Universität zu Berlin, 12203 Berlin, Germany; 7MoKi Analytics GmbH, 12207 Berlin, Germany; 8German Konsiliarlabor for Tropheryma whipplei, 10117 Berlin, Germany; 9Moter Diagnostics, 12207 Berlin, Germany

**Keywords:** Whipple’s disease, *Tropheryma whipplei*, barrier function, endocytosis, invasion, apoptosis, caveolin, Ussing chamber, gastroenteritis

## Abstract

Background: *Tropheryma whipplei* (*TW*) can cause different pathologies, e.g., Whipple’s disease and transient gastroenteritis. The mechanism by which the bacteria pass the intestinal epithelial barrier, and the mechanism of *TW*-induced gastroenteritis are currently unknown. Methods: Using ex vivo disease models comprising human duodenal mucosa exposed to *TW* in Ussing chambers, various intestinal epithelial cell (IEC) cultures exposed to *TW* and a macrophage/IEC coculture model served to characterize endocytic uptake mechanisms and barrier function. Results: *TW* exposed ex vivo to human small intestinal mucosae is capable of autonomously entering IECs, thereby invading the mucosa. Using dominant-negative mutants, *TW* uptake was shown to be dynamin- and caveolin-dependent but independent of clathrin-mediated endocytosis. Complementary inhibitor experiments suggested a role for the activation of the Ras/Rac1 pathway and actin polymerization. *TW*-invaded IECs underwent apoptosis, thereby causing an epithelial barrier defect, and were subsequently subject to phagocytosis by macrophages. Conclusions: *TW* enters epithelia via an actin-, dynamin-, caveolin-, and Ras-Rac1-dependent endocytosis mechanism and consecutively causes IEC apoptosis primarily in IECs invaded by multiple *TW* bacteria. This results in a barrier leak. Moreover, we propose that *TW*-packed IECs can be subject to phagocytic uptake by macrophages, thereby opening a potential entry point of *TW* into intestinal macrophages.

## 1. Introduction

The Gram-positive rod *Tropheryma whipplei* (*TW*) can cause two different pathologies that affect the gastrointestinal tract. The first, classical Whipple’s disease (WD), is a rare, systemic, chronic infectious disease that occurs in only a minority of patients exposed to *TW* and depends on a defect of the host’s innate immune system that is thus far only partially understood [1,2,3,4,5,6]. The second is a self-limiting gastrointestinal infection that makes this bacterium more prevalent in human disease than previously appreciated [7,8,9,10,11,12,13]. While the development of WD undoubtedly requires invasion by *TW*—likely through the mucosal layer—it is currently unknown whether a transient invasion occurs in self-limiting gastrointestinal disease. *TW* is presumed to cause the classical pathology in WD by entering the small intestinal lamina propria (LP), where it afflicts mucosal antigen-presenting cells (macrophages and dendritic cells). However, the mechanism by which the bacteria pass the intestinal epithelial barrier is currently unknown [1,2]. Similarly, the direct trophic effects of *TW* on exposed intestinal epithelia are unknown. This likely reflects the difficulties associated with culturing *TW* [14,15]. Translocation studies have been performed for various other bacteria that pass the intestinal barrier, including *Salmonella*, *Escherichia*, *Shigella*, *Yersinia*, *Campylobacter*, *Pseudomonas,* and *Vibrio* species, revealing distinct transepithelial passage strategies involving M-cell-dependent transcytosis, phagocytosis as well as pinocytosis by intestinal epithelial cells (IECs) [16,17,18,19,20].

Thus, the following study explores (i) the ability of *TW* to pass through the epithelial layer autonomously (directly via IECs, without the help of LP immune cells), (ii) the pathway *TW* utilizes to pass the epithelial barrier, and (iii) the capacity of *TW* to modulate the integrity of the epithelial barrier. The picture arising from the experiments herein suggests that *TW* uptake by IECs and *TW* interference with the intestinal barrier are closely linked and affect each other.

## 2. Results

### 2.1. Establishment of Epithelial Models for Translocation of T. whipplei

It is well-known that in WD, *TW* is found in high numbers within small intestinal LP macrophages. This can be demonstrated either by indirect immunofluorescence using an antibody recognizing *TW* or by PAS staining of the duodenal mucosa of patients with established WD (Figure 1A,B). To determine whether this is secondary to transepithelial bacterial passage and to further elucidate the underlying mechanisms as well as the effects exerted on the intestinal barrier, we sought to establish models that could be used to improve our understanding of epithelial *TW* transmigration. First, we mounted duodenal mucosa from endoscopic biopsies of patients with a healthy duodenum to miniaturized Ussing chambers as previously described and exposed the small intestine by adding a high infectious dose of *TW* (10^8^ bacteria per chamber) to the luminal Ussing compartment for up to 6 h [21]. Although the time of exposure was short compared to the presumed pathophysiology of WD, in which *TW* might be present close to the small intestinal mucosa for much longer periods of time, microscopy revealed sporadic *TW* rods by immunostaining as well as PAS staining (Figure 1C–F). The number of stained rods was considerably lower in the samples exposed to *TW* ex vivo (Ussing chamber) compared to the mucosa of patients with established WD (Figure 1C–F vs. Figure 1A,B). This was an expected finding, given the relatively short period of *TW* exposure in the Ussing experiment compared to the length of exposure in an individual infected with *TW*. Interestingly, the rods were localized to LP-antigen-presenting cells (HLA-DR-positive) that were either CD68-positive macrophages or DC-SIGN-, CD11c-, and S100-positive dendritic cells (Figure 1E,F). Second, we established a cell-culture-based model by apical infection of IEC lines with *TW* (Figure 1G–J). Beginning 3 to 6 h after infection and increasing over time (maximum length of experiment 24 h), immunostained *TW* rods were identified within IECs. Of note, the pattern of infection was heterogeneous, with clusters of *TW*-positive IECs found in between a large number of *TW*-uninfected IECs. IECs infected with a high bacterial load were frequently subject to apical cell shedding (Figure 1H).

### 2.2. Mechanism of Transepithelial Passage of T. whipplei

To obtain a better understanding of the mechanism *TW* utilizes to pass the epithelial barrier, we aimed to inhibit the uptake of *TW* by IECs. Since actin facilitates all types of endocytosis, we applied the inhibitors of actin polymerization latrunculin and cytochalasin D, which reduced *TW* uptake by 59% and 50%, respectively (Figure 2A and Appendix A). Accordingly, aggregations of F-actin were visualized around internalized *TW* (Figure 2B). Moreover, genistein, an inhibitor of tyrosine kinases that was previously shown to inhibit caveolin-dependent endocytosis did not reduce bacterial uptake (Figure 2A and Appendix A) [22]. In a next step, we aimed to specifically block distinct endocytic pathways that may contribute to *TW* uptake via the expression of dominant-negative mutants. The expression of the GTPase-deficient dynamin mutant K44A substantially reduced *TW* uptake by IECs, suggesting a prominent role for the large GTPase dynamin in this process (Figure 2C and Appendix A). In contrast, the expression of a dominant-negative Eps mutant (GFP-ED95/295-Eps) that targets clathrin-dependent endocytosis did not result in significant changes in *TW* uptake (Figure 2C and Appendix A) [22,23]. Subcellular fractionation by ultracentrifugation revealed *TW* was localized to caveolin-1- and caveolin-2-positive fractions (Appendix A). Furthermore, *TW* was found within fractions that were positive for markers of early endosomal (rab5), late endosomal (rab7), and lysosomal (cathepsin) compartments (Appendix A).

Next, we transfected two distinct C-terminal caveolin fragments in IECs: D82-caveolin lacks the capacity to activate Ras, and K135-caveolin is a strong inhibitor of caveolin-based endocytosis (Appendix A) [24]. Interestingly, only the expression of D82-caveolin was sufficient to block *TW* endocytosis, arguing for a role of a Ras GTPase in the endocytic uptake of *TW* and against a direct uptake by caveolae as found for SV40 endocytosis, as this would be inhibited by the expression of K135-caveolin (Figure 2D) [22]. Furthermore, the immunostaining of endogenous IEC caveolin-1 (Figure 2E) and caveolin-2, as well as live cell experiments visualizing the *TW* uptake process of IECs expressing caveolin-mRFP, revealed only rare colocalization events of caveolin and *TW* (Figure 2E and Appendix A). These colocalizations were mostly found in relatively large perinuclear organelles and never in membrane-associated caveolin foci, suggesting that the initial uptake did not occur through classical caveolae. Since caveolin-based uptake has previously been shown to be associated with the activity of small GTPases, cells were pretreated with the Ras inhibitor FTS and the Rac inhibitor EHT-1864, both of which proved to be potent inhibitors of *TW* uptake, further supporting a role for small GTPases in the endocytic uptake of *TW* (Figure 2F and Appendix A) [25]. Cytoskeletal reorganization and actin-driven protrusions, facilitated by the activation of the Ras/Rac1-pathway, are mediated by PI3K [26,27], whose inhibition with LY294002 decreased *TW* uptake. Of note, the caveolin-dependent *TW* uptake by IECs might be subject to positive feedback, as caveolin-1 and caveolin-2 protein levels were found to be upregulated after 24 h of IEC exposition to *TW* (Figure 2G and Appendix A).

When IECs exposed to *TW* were examined by transmission electron microscopy (TEM), intact *TW* rods were identified within *TW*-containing vacuoles that did not fulfil the TEM criteria for caveolae (Figure 3A). Of note, actin polymerization and membrane ruffling occurred at the site of *TW* entry (Figure 3A).

To examine the occurrence of endocytic *TW* uptake in patients with active WD, the duodenal mucosa of WD patients was double-stained for *TW* and endocytic markers. Here, *TW* revealed a robust colocalization with the late endosomal and lysosomal markers Lamp-1 and cathepsin (Figure 3B,C). Of note, endocytic *TW* uptake was restricted to viable bacteria (Appendix A).

### 2.3. Epithelial Barrier in T. whipplei Infection

Next, we aimed to elucidate the effect of *TW* on intestinal barrier function. The transmural electrical resistance (R) was analyzed by mounting healthy human duodenal mucosa on Ussing chambers. Exposure to *TW* caused a moderate (12%), but significant, reduction in R, which was not found in parallel experiments using *E. coli K12* as control bacteria (noninvasive and nonpathogenic bacterium) or when heat-inactivated *TW* were used instead (Figure 4A,D and Appendix A and *E. coli K12*. To identify the mechanism underlying *TW*-associated barrier leaks, we exploited the IEC cell culture model. Importantly, all IEC lines tested showed reduced transepithelial resistance (TER) values upon apical exposure to *TW* (Figure 4B). Most of the cell lines revealed a scatter-like pattern of *TW* uptake. However, HT-29/B6 cells, known to secrete large amounts of apical mucus, revealed patches of numerous apical *TW* rods [28]. To determine whether IEC-secreted mucus might contribute to the capacity of *TW* to adhere to IECs, we biochemically removed the mucus using the secretolytically active drug *N*-acetylcysteine (NAC). Treatment with NAC removed the characteristic *TW* patches from the HT-29/B6 monolayer (Figure 4C,E). Although NAC pretreatment was insufficient to rescue the TER decrease in HT-29/B6 cells, it ameliorated the barrier defect found in human intestinal mucosae mounted on Ussing chambers (Figure 4B,F).

### 2.4. Epithelial Apoptosis Due to T. whipplei Exposure

Next, we hypothesized that the *TW*-triggered apoptosis of IECs might contribute to the barrier defect described above. We thus quantified activated caspase-3 in epithelia exposed to *TW* and found an 11-fold induction, which was abrogated when cells were pretreated with an apoptosis inhibitor (Figure 5A,D). Interestingly, proliferation was decreased upon *TW* exposure (Figure 5B,D). The inhibition of apoptosis ameliorated the barrier defect. However, a substantial portion of the barrier defect appeared to be unrelated to apoptosis since some barrier defect remained although the inhibition of apoptosis appeared to be complete (Figure 5A,C). Interestingly, confocal microscopy as well as TEM revealed that IECs with a high load of *TW* were particularly prone to undergoing apoptosis-associated cell death (Figure 1H and Figure 5E). Furthermore, *TW* uptake was more frequently found in apoptotic IECs compared to nonapoptotic cells (Appendix A). Thus, these results suggested that *TW*-packed IECs in particular undergo apoptosis, thereby contributing to an epithelial barrier defect.

### 2.5. Uptake of Apoptotic, T.-whipplei-Bearing Intestinal Epithelial Cells by Macrophages

Thus far, we have presented evidence that (i) *TW* is taken up by epithelial cells via a distinct endocytic process and (ii) *TW*-bearing IECs are likely to undergo apoptosis, thereby causing a barrier leak. Hypothetically, both processes could together promote the final translocation of *TW* into the subepithelial compartment. *TW*-containing IECs undergoing apoptosis could not only shed into the intestinal lumen but also be subject to phagocytosis by intestinal macrophages. Thus, we used a modified version of our previously published coculture model with Caco-2 cells and in vitro differentiated human macrophages (Appendix A) [29]. Caco-2^Actin^ cells were seeded on filter transwells with macrophages attached to the well bottom. After allowing the Caco-2^Actin^ cells to form an epithelial layer, the cells were exposed to *TW* for 24 h. To elucidate whether *TW* together with IEC (apoptotic) material was taken up by subepithelial macrophages, confocal imaging was performed (Figure 5F and Appendix A). Most macrophages were found to be positive for actin-GFP when seeded next to Caco-2^Actin^ cells, suggesting a regular uptake of IEC material by macrophages (Figure 5F and Appendix A). As a negative control, macrophages incubated with native Caco-2 cells did not reveal intracellular actin-GFP (Appendix A). However, the same macrophages that showed IEC remnants also contained *TW* material when exposed to *TW*. This result revealed the capacity of macrophages to take up *TW* and IECs simultaneously.

## 3. Discussion

WD and *TW*-induced gastroenteritis can develop in individuals exposed to *TW* after the uptake of this bacteria into the intestinal mucosa. Since the mechanisms that enable *TW* to enter the intestinal mucosa and the pathway *TW* utilizes to pass the IEC layer has not yet been characterized, we conducted a study using ex vivo human intestinal mucosa as well as genetically altered IECs to elucidate both the underlying mechanism of *TW* uptake and its impact on the epithelial barrier.

The mucosal barrier is a complex structure that shields the mucosa from the unlimited passage of luminal microorganisms and toxins into the system while regulating the uptake of water, electrolytes, and nutrients. It includes the mucus layer, which contains bicarbonate, mucins, and antimicrobial peptides, and the epithelial barrier, which is determined by the integrity of the IECs, the sealing of the paracellular space by a delicately regulated apical junctional complex, and the activity of the transcellular route, which is capable of transporting macromolecules as well as particles the size of microorganisms [30,31]. It is further regulated by a heterogeneous set of LP cells, including mucosal macrophages and dendritic cells that can, through their interaction with IECs, focally alter barrier function or even collect luminal antigen by extending transepithelial dendrites into the epithelium [29,32,33].

### 3.1. Transcellular Route of T. whipplei

Several pathogenic microorganisms, including *Salmonella*, *Escherichia*, *Shigella*, *Yersinia*, *Campylobacter*, *Pseudomonas*, and *Vibrio* species, have developed strategies to overcome this barrier, thereby invading the human organism. The preferred entry site along the gastrointestinal tract varies with the microorganism of interest, with *Salmonella* preferentially invading Peyer’s patches in the ileal mucosa, *Campylobacter* invading small intestinal and colonic mucosa, and *Shigella* and *Escherichia* species being optimized to enter via the colonic mucosae [34,35,36]. In contrast to these pathogens, in gastrointestinal WD, *TW* is mostly found in small intestinal or duodenal macrophages or dendritic cells. However, infected macrophages can also be detected in the large intestinal mucosa, although in much lower numbers than in the small intestine [2]. Thus, we proposed IECs to be the primary entry site for *TW* bacteria, which was supported by the autonomous entry of *TW* in IECs in vitro, the immunostaining of duodenal tissue of WD patients, and the engulfment of TW by epithelial cells and intestinal macrophages when human duodena were exposed to TW ex vivo.

A further question relates to the entry route bacteria use to trespass the IEC layer: transcellular or paracellular. The immunostaining of human mucosae as well as the immunostaining and TEM of IEC cell lines revealed frequent intracellular *TW* but no paracellular *TW*. Thus, we concluded that *TW* translocated via endocytosis rather than via the facilitated permeabilization of apical junctions. This was further supported by the finding that we could block *TW* uptake by the inhibition of apical endocytosis. Similarly, a number of intestinal bacterial pathogens directly enter IECs, which subsequently pass the microbes on to a subepithelial compartment by either a basal secretion of bacteria or the phagocytosis of affected IECs through subepithelial mononuclear cells [35,37]. In contrast, for some bacteria, including *Campylobacter* and hemolysin-positive *E. coli*, the invasion of the intestinal mucosae was shown to occur via a paracellular pathway, by either a controlled entry through tight junctions or focal leaks [37,38,39]. Although we did not detect any paracellular *TW*, we cannot rule out this route completely, as the effective inhibition of IEC apoptosis only partially reversed the effect on the IECs’ barrier function.

### 3.2. Mechanism of Endocytic Uptake

The next question arising is: What is the molecular mechanism underlying *TW* uptake? Our study suggests a mechanism dependent on the actin cytoskeleton, since latrunculin and cytochalasin D were both able to reduce *TW* uptake, as well as on the large GTPase dynamin, since the expression of a dominant-negative dynamin mutant reduced *TW* uptake. Dynamin is known to pinch off vesicles from the cell membrane in a GTP-dependent manner via its mechanoenzyme activity [40]. Of note, it has been shown that dynamin is upstream of rab5, which is required for early endosome formation [41].The dynamin dependency of bacterial uptake by IECs has previously been shown for *Listeria monocytogenes* and *Salmonella* spp. [42,43]. Furthermore, based on the results of our experiments using the GFP-ED95/295-Eps mutant, we propose that clathrin-coated vesicle formation does not play any role in *TW* uptake. Instead, *TW* endocytosis might be affected by caveolin function, not caveolae as such, which is supported by the analysis of *TW* uptake after the expression of two different caveolin fragments. *TW* endocytosis was not affected by the K135-caveolin fragment known to strongly affect caveolin-mediated endocytosis (which is not surprising, since TEM imaging showed no apical surface caveolae and *TW* is far too large for endocytosis by a single caveola) but was impaired by the expression of the longer D82-caveolin fragment, which selectively impairs the activation of membrane-associated Ras [24]. Thus, the distinct function of caveolin might be to activate signaling via small GTPases, which then act on the actin cytoskeleton [44,45]. Further evidence for this interpretation results from experiments using two inhibitors of small GTPases (FTS and EHT-1864) that reduced *TW* endocytosis to a significant extent and from the experiments using actin inhibitors. Previously published specificity data of these GTPase inhibitors suggested that Rac1 rather than Ras GTPase was actually inhibited [46,47]. Interestingly, it has been shown only recently that Rac1 activity must be tightly controlled during the process of macropinocytosis and that this process is preceded by the activation of caveolin-1 [48,49]. Moreover, the activation of the small GTPase Rac1 by caveolin has previously been shown to play a role in the invasion of *Salmonella typhimurium* in a HeLa cell model [25]. Therefore, our data in combination with previously published data suggest a mechanism that includes the activation of Ras and Rac1 (and, subsequently, dynamin), with caveolin acting as a scaffold in the IEC membrane (Figure 6).

Within macrophages, *TW* creates a niche by manipulating steps of phagosome–lysosome maturation, thereby facilitating its replication in an acidic environment, which is critical for its prolonged persistence in host cells during WD [50,51,52]. We suggest a similar mechanism in IECs, since several proteins derived from the endolysosomal pathway are enriched in the *TW*-containing vacuole. Our data suggest that the passage of *TW* through IECs seems to be related to phagocytosis or macropinocytosis.

The focus of the current study was to identify host-related mechanisms that allow *TW* to pass the human intestinal barrier. One further exciting question not answered by this study relates to virulence factors *TW* might use to trigger its own uptake by the IEC layer. The sequencing of the *TW* genome has uncovered two genes, *TWT_701* and *TWT_189*, that were grouped as putative type-II/type-IV pathway secretion systems (T4SS) by a conserved domain analysis [53]. Notably, *TW* has a reduced genome and type-II/type-IV secretion systems are complex and large protein complexes [53]. Type-IV secretion systems have been identified in *Helicobacter pylori*, *Neisseria meningitides*, and *Pseudomonas aeruginosa* and consist of pili through which bacteria insert proteins into the host cell. Interestingly, it was shown at a molecular level that *Neisseria* passed the blood–brain barrier by inducing a T4SS-dependent switch in endothelial cell polarity [54]. Similarly, the passage of *Pseudomonas aeruginosa* through an IEC layer is facilitated by type-IV pili [55]. Thus, the two *TWT* genes might be an interesting starting point for future research on pathogen-related factors facilitating the *TW* penetration of the intestinal epithelium.

### 3.3. Barrier Defect Caused by T. whipplei

Another question to address was the integrity of the small intestinal mucosa and the IEC layer following contact with and entry of *TW*. By analyzing the ex vivo Whipple model and by conducting TER studies on various IEC cell lines, we presented evidence that *TW* impaired epithelial barrier function via its potential to induce IEC apoptosis. Notably, this might also explain our observation that inhibition of endocytic pathways not completely prevented the transepithelial passage of *TW*. It has been shown that *TW* also induces apoptosis in macrophages via caspase-3 [56]. Naturally, a reduction of barrier function has been found for other pathogens, including enteropathogenic *E. coli* (EPEC), *Campylobacter*, and *Yersinia enterocolitica* [57,58,59]. The barrier defect described in our ex vivo model occurred after a short exposure (hours) of intestinal epithelia to *TW* and thus might not reflect the mechanism of diarrhea found in WD [60,61]. However, since the finding suggests that the barrier defect might result in a leak flux, it might play an important role in *TW*-induced gastroenteritis [7]. Leak flux diarrhea has previously been identified as a causative mechanism in various enteropathies, including celiac disease, lambliasis, norovirus infection, and HIV enteropathy [62]. Interestingly, we found some evidence for the mucus layer having a role in facilitating *TW* adherence to the epithelial layer and a consequent reduction in barrier function. Similarly, *Campylobacter jejuni* was shown to adhere to intestinal mucus and even propagate within the mucus, as well as to impair barrier function [63]. This highlights that the ability of intestinal pathogens to interact with the mucus layer is an independent determinant for pathogen virulence.

Is the modulation of barrier function also related to the *TW* uptake mechanism discussed above? It has previously been proposed that pathogens entering the intestine might invade the mucosa primarily by entering IECs, which then become apoptotic and are phagocytosed by LP monocytic cells. This might be of specific relevance to WD, since its hallmark is the occurrence of mucosal macrophages harboring *TW*. Applying *TW* on the coculture model that we had previously established suggests that this might in fact hold true for *TW* uptake as well, since both epithelial and *TW* components were found to be enriched in the subepithelial macrophages [29]. However, further studies will be necessary to further characterize this specific aspect.

The transmission of *TW* via the fecal–oral route is likely. However, the ability of *TW* and the mechanism by which the bacteria pass the intestinal epithelial barrier are currently unknown. Hence, we used static models (with inherent limitations) to study selected aspects of *TW* infection.

In summary, we propose a transepithelial uptake mechanism for *TW* into the small intestinal mucosa that includes dynamin- and caveolin-dependent, but not clathrin-dependent, endocytic mechanisms. *TW*-harboring IECs might undergo apoptosis, which then leads to uptake by LP macrophages and a defective epithelial barrier (Figure 6). The latter presumably contributes to *TW*-associated diarrhea by a leak flux mechanism.

## 4. Materials and Methods

### 4.1. Chemicals

EHT 1864 and farnesylthiosalicylate (FTS) were obtained from Santa Cruz (Dallas, TX, USA). Latrunculin was purchased from Calbiochem (Darmstadt, Germany), LY294002 from Cell Signaling (Cambridge, UK), and cytochalasin D and genistein from Sigma-Aldrich Chemicals (Munich, Germany). Z-VAD-FMK was obtained from Promega (Mannheim, Germany) and *N*-acetylcysteine (NAC) from Hexal (Holzkirchen, Germany).

### 4.2. Constructs

Human wild-type and K44A-dynamin1 were subcloned from pcDNA3 constructs into pEGFPN1 via the EcoR1 and BamH1 sites [64]. GFP-fused wild-type and GFP-ED95/295 mutant Eps constructs were a gift from Alexandre Benmerah (Université Paris Descartes, France) [23]. Caveolin1-mRFP and -GFP constructs were a gift from Lucas Pelkmans (University of Zürich, Switzerland) [22]. D82- and K135-caveolin constructs were generated in analogy to the study by Roy et al. using a PCR-based strategy and subcloned into an mRFP-modified pEGFPN1 vector via EcoR1 and Mfe1 sites (Appendix A) [24]. When caveolin mRFP fragments were expressed together with wild-type caveolin-EGFP, distinct differences in cellular distribution and motility were found, as previously described (Appendix A) [24].

### 4.3. Bacterial Strains and Preparations

*T. whipplei* Twist Marseille (CNCM I-2202) was cultured in axenic medium as described previously and used as viable or heat-killed bacteria (100 °C, 90 min) [14,51,56]. Prior to infection, bacteria were resuspended in cell-line-specific medium.

### 4.4. Cell Culture and Transfection

The human colon carcinoma cell lines T84, Caco-2, and HT-29/B6 and the porcine jejunal cell line IPEC-J2 (all from ATCC) were grown on filter inserts (T84 and HT-29/B6: PCF, 3 µm; Caco-2: PCF 0.4 µm; IPEC-J2: HA 0.45 µm; Merck Millipore, Darmstadt, Germany) in DMEM/F-12 Ham containing 10% fetal calf serum (FCS; Sigma-Aldrich Chemicals), MEM with GlutaMax (15% FCS; Gibco Life Technologies), RPMI-1640 (10% FCS; Sigma-Aldrich Chemicals), and DMEM/F-12 Ham with 5% porcine serum, respectively. Media were complemented with antibiotics (100 U/mL penicillin and 0.1 mg/mL streptomycin; Sigma-Aldrich Chemicals). IEC monolayers were apically inoculated with *TW*, *E. coli* K12, or a control medium for 24 h (multiplicity of infection (MOI) = 100) starting on culture day 12 (T84), 14 (Caco-2), 7 (HT-29/B6), and 7 (IPEC-J2), respectively. The MOI refers to the counted and calculated cell number directly before the infection step. *E. coli* K12 was used as a control because in the *E. coli* K12 strain, most virulence factor genes that are present in intestinal and extraintestinal pathogenic *E. coli* strains are absent. Furthermore, *E. coli* K12 is unable to attach to the mucosa of the intestinal epithelium. A barrier defect (as expressed by a TER reduction) has not been described. Transepithelial resistance (TER, Ω × cm^2^) was measured before and 24 h after the addition of *TW* using an ohmmeter (D. Sorgenfrei, Institute of Clinical Physiology, Berlin, Germany). Glucose level and pH (Merck Millipore) were monitored to exclude bacteria-induced medium consumption. No significant differences in pH or glucose concentration were detected (pH (mean ± SD): control 7.5 ± 0.0, *TW* 7.5 ± 0.2; glucose concentration (µM; mean ± SD): control 3.65 ± 1.05, *TW* 3.09 ± 0.94). For infection with viable bacteria, monolayers were shifted to their respective antibiotic-free media 24 h before experiments. Cell apoptosis inhibition was achieved by a treatment with the pan-caspase inhibitor Z-VAD-FMK (50 µM). Caco-2 cells stably transfected with Actin-GFP (Caco-2^Actin^) were a generous gift from Jerrold R. Turner (Departments of Pathology and Medicine [GI], Brigham and Women’s Hospital and Harvard Medical School, Boston, MA, USA).

### 4.5. Inhibition of Endocytic T. whipplei Uptake

For the inhibition of *TW* uptake, latrunculin A (0.2 µM), cytochalasin D (20 µM), and genistein (25 µM), as well as the Rac-GTPase inhibitor EHT 1864 (5 µM), the Ras-GTPase inhibitor FTS (20 µM), and the phosphoinositide 3-kinase (PI3K) inhibitor LY294002 (25 µM) were added to the apical compartment. For secretolysis, NAC (10 mM) was added to the apical compartment of HT-29/B6 cells. Cells transiently transfected with constructs containing dominant-negative endocytosis mutants coupled to EGFP were exposed to *TW*. The transient transfection of Caco-2 was performed using the Amaxa Nucleofector and Kit T (Lonza, Basel, Switzerland) according to the manufacturer’s instructions. The uptake of *TW* by Caco-2 cells was assessed by flow cytometry and expressed as the mean fluorescence intensity normalized to respective controls (Appendix A). Trypsinized and filtered cells were washed with phosphate-buffered saline containing 10 mM EDTA and 0.5% bovine serum albumin to remove *TW* attached to the cell surface, fixed in paraformaldehyde, and permeabilized with 0.5% Triton-X and saponin before staining. IEC cell suspension was analyzed by flow cytometry with rabbit-anti-*TW* (provided by Didier Raoult, Institut Hospitalo-universitaire Méditerranée Infection, Marseille, France; Aix Marseille Université, Institut de Recherche pour le Développement, Assistance Publique-Hôpitaux de Marseille, Microbes Evolution Phylogénie et Infections, Marseille, France), and visualized using the appropriate secondary donkey antirabbit antibody linked to Alexa488 or Alexa 647 (Life Technologies, 1:500). Data were acquired on a FACSCalibur or FACSCanto II (both BD Biosciences) and analyzed with FlowJo software (FlowJo, Ashland, OR) (for the gating strategy, see Appendix A). The *TW*-specific antibody is validated and used for routine clinical diagnostics [15,65]. The monolayer experiments have revealed that IECs are infected infrequently together with a heterogeneous pattern of *TW*-infected cells after 24 h. Only few cells engulf *TW* and the amount of *TW* per cells varies markedly. This heterogeneity, together with the infrequency, explains that no clear distinct *TW* infected population can be separated.

### 4.6. T. whipplei Translocation in a Caco-2/Macrophage Coculture Model

Monocytes were isolated from human peripheral blood using CD14 magnetic cell sorting (MACS, Miltenyi Biotech, Bergisch-Gladbach, Germany) and differentiated in vitro to macrophages by the addition of granulocyte-macrophage colony-stimulating factor (500 U/mL; Bayer, Berlin, Germany) to generate M1 macrophages, as previously published [29]. Hanging filter inserts (Nunc) were placed in 12-well plates with the lower filter side facing up. Caco-2^Actin^ cells (and for controls, nontransfected Caco-2 cells) were seeded on the lower filter side and were allowed to attach for 8 h (200,000 cells per filter insert, MEM with GlutaMax). Filter inserts were then transferred to a 24-well plate (now with the upper filter side facing up) to seed macrophages (1.5 × 10^5^ cells per filter inserts) (Appendix A). On culture day 14, *TW* was added to the apical compartment (MOI = 200) (= infection via the basolateral side). After 24 h of *TW* exposure, filters were washed and subjected to fixation (4% PFA, 15 min) and permeabilization (0.5% Triton X-100, 5 min). Cytospins of the cells of the macrophage compartment were prepared with a Cytospin 2 Cytofuge (Thermo Shandon, Frankfurt, Germany) at 100× *g* for 7 min, air-dried, fixed in 1% PFA for 60 min at 4 °C, and stored at −80 °C.

### 4.7. Immunostaining and Confocal Microscopy

Filters were washed twice with PBS^+^ (PBS containing Ca^2+^ and Mg^2+^) and fixed with 2% paraformaldehyde (PFA) in PBS^+^ for 15 min. After completion of the Ussing experiments, biopsies were unmounted and fixed with 4% PFA (6 h, RT). Immunostaining was performed on paraffin sections of duodenum as previously described [66]. Primary antibody against cleaved caspase-3 (Asp175; Cell Signaling Technology, 1:600), anti-*TW*, anti-CD68 (PG-M1, 1:200), anti-Ki-67 (MIB-1, 1:500), anti-HLA-DR (DakoCytomation, 1:500), anti-CD11c (5D11, Leica Biosystems, 1:50), anti-DC-SIGN/CD209 (111H2.02, Dendritics, 1:150), anti-S100β (EP1576Y, Millipore, 1:250), anti-Lamp-1 (Santa Cruz Biotechnology, 1:50), and anti-E-cadherin (1:500) and anti-Cathepsin D (both Abcam, 1:200) were visualized using the appropriate secondary antibody linked to either Alexa555/594 or Alexa488 (Life Technologies, 1:500) together with actin staining using Alexa647-Phalloidin (Life Technologies, 1:500) [66]. Nuclei were counterstained with DAPI (Sigma, 1:1000). Periodic acid-Schiff (PAS) staining was done by routine pathological diagnostics. Apoptosis (caspase-3-positive) and proliferation index (Ki-67-positive) were expressed as the percentage of cleaved caspase3-positive cells and Ki-67-positive cells, relative to the total number of epithelial cells per filter. Fluorescence was detected using an LSM 510 Meta laser scanning microscope (Zeiss, Jena, Germany), and image processing was performed using LSM software AIM.

### 4.8. Electron Microscopy

Cell suspensions were fixed overnight in 2.5% glutaraldehyde in 0.1 M cacodylate buffer. Samples were washed, postfixed with 1% OsO_4_ and 0.8% K_4_(Fe(CN)_6_) in 0.1 M cacodylate buffer for 1.5 h, dehydrated with graded ethanol solutions, and embedded in Epon (SERVA).

Semithin sections were stained with Richardson’s solution. Ultrathin sections were stained with uranyl acetate and lead citrate and analyzed using a Zeiss EM 906 electron microscope.

### 4.9. Live Cell Microscopy

Caco-2 cells were grown on Lab-Teks (8-well-Lab-Tek™ II chambered coverglass, ThermoScientific) and transfected with either caveolin1-EGFP or D82- and K135-caveolin-mRFP fragments, as stated above. *TW* exposure was performed in a Hepes Ringer solution (pH 7.4, with the addition of 10 mM D-glucose) at an MOI of 100. *TW* was visualized via in vivo staining (BacLight stain-488, Molecular Probes). Live cell imaging was performed during the 12 h following *TW* exposure using a Zeiss LSM 780 equipped with a PM S1-Incubator controlled by a computer-assisted heating system (Incubation system S, Zeiss).

### 4.10. Analysis of Barrier Function of Small Intestinal Mucosae

Duodenal biopsy specimens of individuals without clinical signs of enteropathy were obtained during routine endoscopy from 8 patients (5 males, 3 females; age range 28–75 years, mean 54.7 years). Mucosae were mounted to the miniaturized Ussing chamber system as previously described [21,67]. Resistance values determined by a computerized automatic clamp device (Fiebig Hardware & Software, Berlin, Germany) were corrected for the resistance of the bathing fluid between the voltage-sensing electrodes and the empty filter. As a bathing solution, 10 mL of DMEM/F-12 Ham was used per chamber side. After 30 min of equilibration, bacteria (10^7^ per chamber) were added to the mucosal compartment and resistance was monitored for up to 3 h. To reduce mucus levels, mucosae were pretreated with NAC (10 mM) for 15 min. Biopsies from patients with WD were obtained at the time of diagnosis.

### 4.11. Purification of the Intracellular Compartment Containing T. whipplei

For fractionation, Caco-2 cells that had been exposed to *TW* were lysed in 500 mM Na_2_CO_3_, Dounce homogenized (Wheaton Science, 20 strokes on ice), and homogenized further using an Ultra Turrax (3 × 10 s) [68]. A sucrose density gradient was established in a 12 mL Beckman ultracentrifugation tube by mixing 2 mL of the homogenate with 2 mL of 90% sucrose, followed by applying 4 mL 35% sucrose and then 4 mL 5% sucrose on top. Ultracentrifugation was performed at 54,000 rpm (20 h, 4 °C, Beckman’s T70.1Ti-rotor), which is equivalent to a maximum relative centrifugal force of approx. 240,000× *g*. After centrifugation, 500 µL fractions were carefully collected to be later analyzed by Western blotting.

### 4.12. Western Blotting

Immunoblots were analyzed as described elsewhere [21]. Detergent-soluble protein fractions were prepared from Caco-2 monolayers incubated with *TW* or controls and detected by the primary monoclonal anti-Caveolin-1 (Sigma-Aldrich), anti-Caveolin 2 (BD Pharmingen, Heidelberg), anti-rab5, anti-rab7, and anti-cathepsin (Cell Signaling, Cambridge), and anti-*TW* antibodies.

### 4.13. Statistical Analysis

Results were expressed as median, and the dispersion of data is expressed as first and third quartiles with minimum and maximum. All analyses were performed using GraphPad Prism version 9.3.1 software. After performing normality testing, single comparisons were assessed using either a two-tailed, unadjusted, unpaired Student’s *t*-test or a Mann–Whitney test. Differences among groups were analyzed with an analysis of variance, followed by the Bonferroni adjusted t-test or Kruskal–Wallis test with Dunn’s test to correct for multiple comparisons. *p*-values <0.05 were considered statistically significant.

## 5. Conclusions

*TW* enters IECs via an actin-, dynamin-, caveolin- and Ras-Rac1-dependent endocytosis mechanism and consecutively causes IEC apoptosis primarily in IECs invaded by multiple TW bacteria. This results in a barrier leak. Moreover, we propose that *TW*-packed IECs can be subject to phagocytic uptake by macrophages, thereby opening a potential entry point of *TW* into intestinal macrophages. Insufficient bacterial clearance in a predisposed host will cause WD and more frequently, transient gastroenteritis.

## Figures and Tables

**Figure 1 ijms-24-06197-f001:**
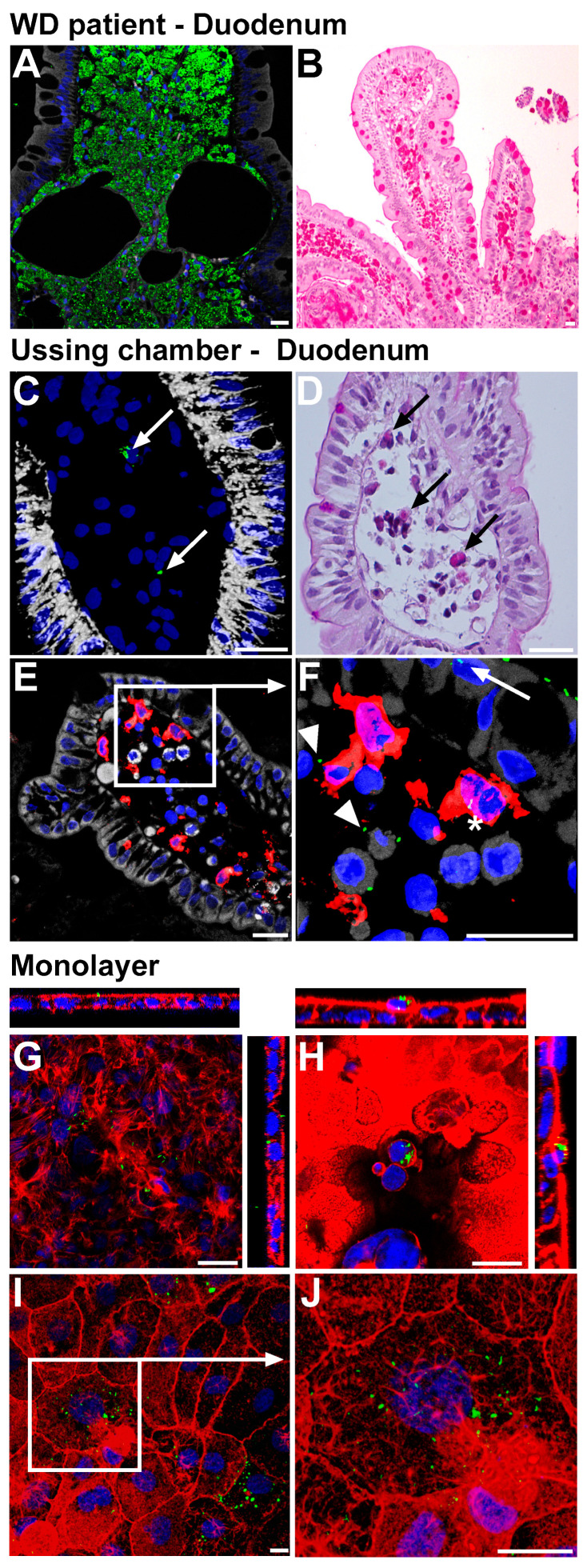
(**A**,**B**): Models for infection of intestinal mucosa/epithelia by *T. whipplei*. Representative section of an untreated WD patient at time of diagnosis, exhibiting a massive infiltration of duodenal lamina propria (LP) macrophages with *TW*. Immunofluorescence staining (IF), green: *TW*, white: F-actin, blue: nuclei (**A**). Periodic acid-Schiff (PAS) staining (**B**). (**C**–**F**): Sections (IF and PAS) of healthy duodenal mucosa that were luminally exposed ex vivo to *TW* in Ussing chambers for 3 h. *TW* was detected within LP (arrows in (**C**,**D**)). Counterstaining for epithelia ((**E**)—cadherin, white) and antigen-presenting LP cells (HLA-DR, red). PAS-positive material within LP cells (arrows, (**D**)). *TW* (green) within the epithelial layer (arrow), within HLA-DR-positive cells (asterisk), and intercellularly within the LP (arrowhead). (**F**): detail of (**E**). (**G**–**J**): IF-stained sections with single confocal XY-planes of Caco-2 (**G**,**H**) and IPEC monolayers ((**I**), detailed in (**J**)) that were apically exposed to *TW* for 24 h. Red: Cytoskeletal F-actin. *TW* was found intra- and subepithelially (**F**) and accumulated in rounded cells that were shed to the apical compartment (**H**). Size bars, 20 µm (**A**–**F**), 10 µm (**G**–**J**).

**Figure 2 ijms-24-06197-f002:**
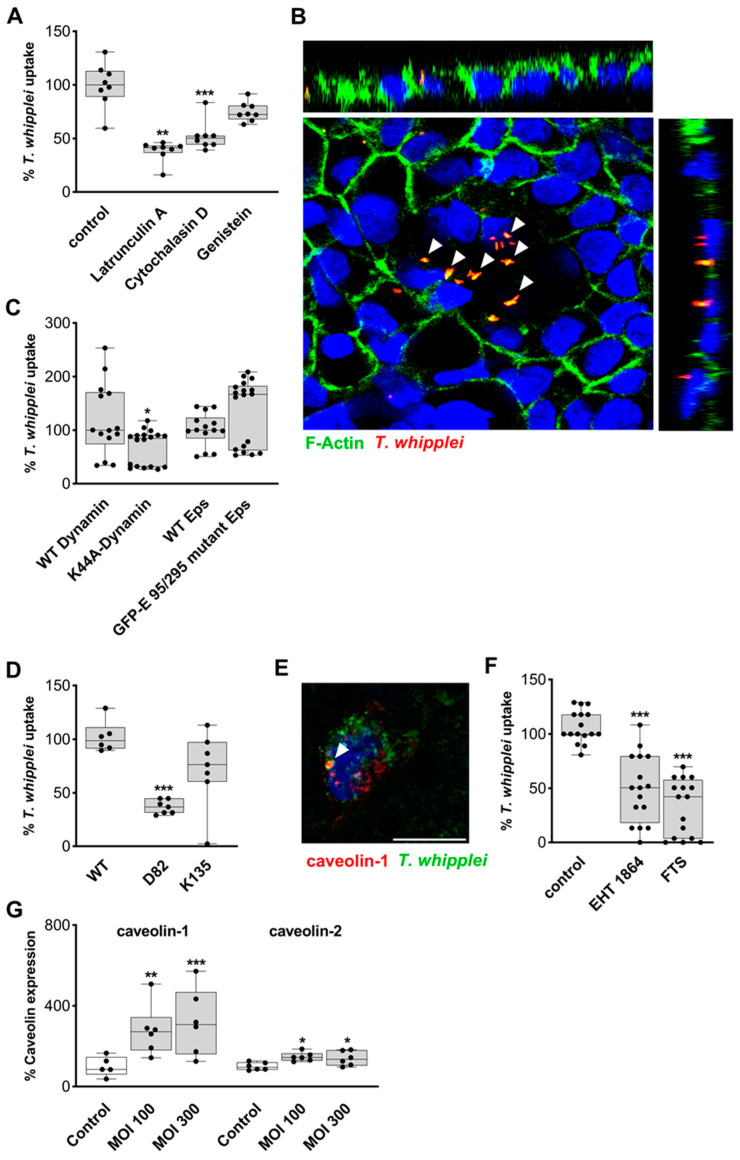
Functional inhibition of *T. whipplei* uptake. Caco-2 cells were inoculated with *TW* for 24 h (multiplicity of infection = 100). (**A**): Monolayers of Caco-2^Actin^ after apical inoculation with *TW*. Epithelial uptake of bacteria was assessed by flow cytometry. Cells were treated with inhibitors of actin polymerization (latrunculin A, cytochalasin D) and genistein. The percentage of *TW* uptake was determined (*n* = 8). (**B**): Immunofluorescence/confocal imaging of Caco-2^Actin^ cells exposed to *TW*. *TW* (red) co-localizes with GFP-actin (green); colocalization = intense yellow (indicated by arrowheads). Nuclei are stained in blue. (**C**,**D**): Bacterial uptake after inhibition of the large GTPase dynamin- (K44A), clathrin- (ED95/295 mutant Eps, C; *n* = 14–18), and caveolin-dependent (D82, K135, D; *n* = 6–7) endocytic pathways by the expression of dominant-negative mutants and their respective wild-type constructs. (**E**): Immunofluorescence/confocal imaging of *TW* (green) and endogenous caveolin (red) of Caco-2 cells exposed to *TW*. (**F**): Inhibition of *TW* uptake in Caco-2 cells by inhibitors of small GTPases: FTS (20 µM) and EHT-1864 (5 µM; *n* = 16). (**G**): Quantification of Western blot of total lysates for caveolin-1 and caveolin-2 after exposure of Caco-2 cells to *TW* (*n* = 5–6). (**A**,**C**,**D**,**F**,**G**): Results are expressed as single values with median. Dispersion is expressed as minimum to maximum. Significance levels were corrected for multiple comparisons. * *p* < 0.05, ** *p* < 0.01, *** *p* < 0.001. Size bars, 20 µm.

**Figure 3 ijms-24-06197-f003:**
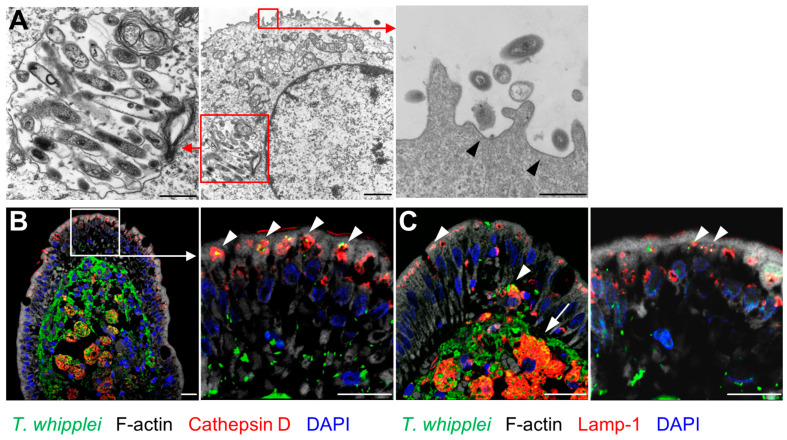
Compartments involved in *T. whipplei* endocytosis. (**A**): T84 cell after apical uptake of *TW* analyzed by transmission electron microscopy. Large vesicles containing numerous intact *TW* rods (left panel). Formation of *TW*-containing vesicles with apical membrane condensation and protrusions (right panel, arrowhead). Size bars, middle panel: 1 µm; left and right panel: 500 nm. (**B**,**C**): Compartments involved in *TW* endocytosis in untreated WD patients. Details of the epithelial layer are shown in the right panels. (**B**): Immunofluorescence staining of *TW* (green), cathepsin D (red), F-actin (white), and nuclei (blue). *TW* is located in an apical lysosomal compartment (red, cathepsin D) within the epithelium (arrowhead). (**C**): *TW* is located in an intraepithelial late endosomal compartment (Lamp-1, red; arrowhead) and within lamina propria macrophages (arrow). Size bars, 20 µm.

**Figure 4 ijms-24-06197-f004:**
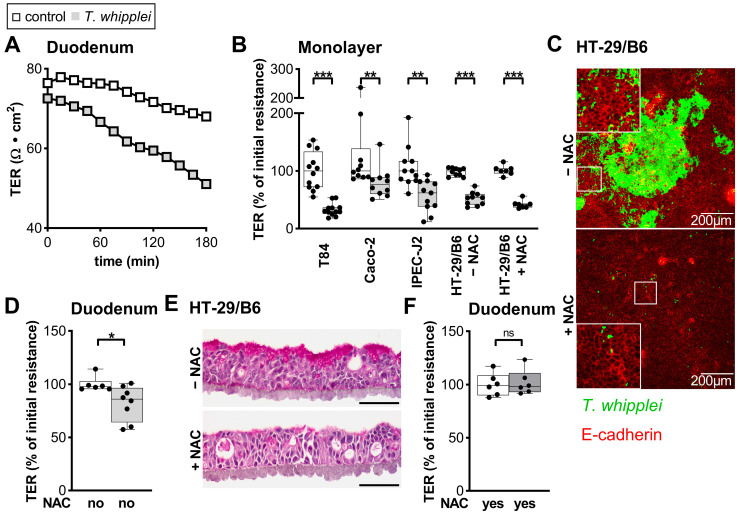
*T. whipplei* interrupts epithelial barrier. (**A**): Representative time course of barrier defect of human duodenal mucosa inoculated ex vivo with *TW*. (**B**,**D**): *TW*-dependent decrease in transepithelial resistance (TER) in T84, Caco-2, IPEC-J2, and HT-29/B6 monolayers (24 h inoculation, multiplicity of infection = 100) and human duodenal mucosa (3 h inoculation, 10^7^ bacteria per Ussing chamber). Graphs depicting single values, median, and minimum to maximum, *n* = 6–12, * *p* < 0.05, ** *p* < 0.01, *** *p* < 0.001. (**C**–**F**): Role of the mucus layer in *TW*-induced barrier defect. Pretreatment of HT-29/B6 with *N*-acetylcysteine (NAC). (**C**): Immunofluorescence staining (performed after 24 h inoculation and washing steps). NAC reduces apical adherent *TW* which remains after performing apical washing steps. *TW*, green; E-cadherin, red. Size bars, 200 µm. (**E**): Periodic acid-Schiff staining of HT-29/B6 monolayers reveals a reduced apical mucus layer after treatment with 10 mM NAC. Size bars, 20 µm. (**F**): NAC pretreatment prevents epithelial barrier defect by *TW*, as revealed by TER measurement in Ussing chambers 3 h after *TW* inoculation. Notably, the apical fluid compartment recirculates compared to the static conditions during monolayer experiments. Bars represent single values, median, and minimum to maximum, *n* = 6, ns: not significant.

**Figure 5 ijms-24-06197-f005:**
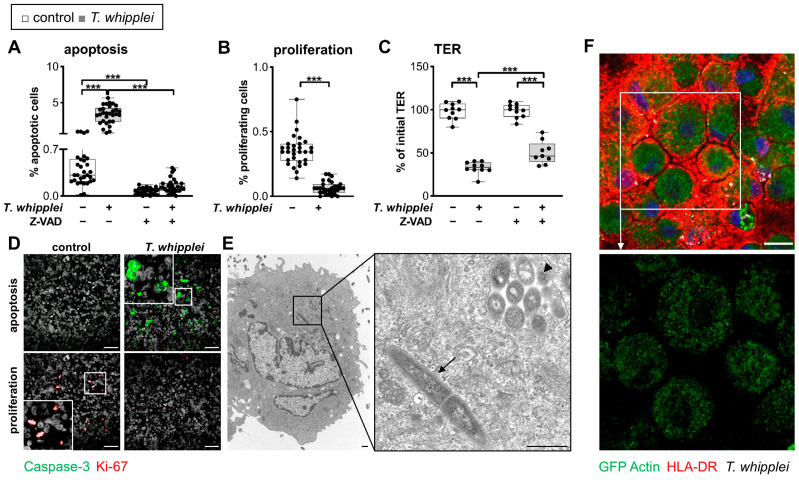
*T. whipplei* induces epithelial apoptosis. (**A**): Percentage of cleaved caspase-3-positive T84 cells by immunofluorescence (IF) microscopy 24 h after inoculation with *TW* (multiplicity of infection (MOI) = 100) ± caspase inhibitor ZVAD (50 mM). (**B**): Percentage of Ki-67-positive T84 cells by IF microscopy 24 h after inoculation with *TW* (MOI = 100). Apoptosis (caspase-3-positive) and proliferation index (Ki-67-positive) were expressed as the percentage of cleaved-caspase3-positive cells and Ki-67-positive cells, relative to the total number of epithelial cells per filter. (**C**): Relevance of *TW*-induced apoptosis for barrier function. Transepithelial resistance (TER) was measured 24 h after *TW* inoculation ± ZVAD. Bars represent single values, median, and minimum to maximum, *n* = 9–30. Significance levels were corrected for multiple comparisons. *** *p* < 0.001. (**D**): Representative confocal IF images (green: caspase-3, red: Ki-67, white: DAPI). Size bars, 50 µm. (**E**): Transmission electron microscope image of a detached apoptotic T84 cell revealing intact intracytosolic (arrow) and intravesicular (arrowhead) *TW* rods. Detached cells were obtained from the apical supernatant of a T84 epithelial monolayer. Criteria for apoptosis: rounded, single cell, cell shrinkage, and nuclear fragments. Size bars, 0.8 µm. (**F**): Confocal IF images of *TW* passage into the macrophage compartment through the epithelial layer, as revealed by the coculture model of Caco-2 cells and primary human macrophages. Macrophages attached to the well bottom are HLA-DR-positive (red) and contain epithelial actin-GFP (green) as well as *TW*-positive material (white). Size bar, 20 µm.

**Figure 6 ijms-24-06197-f006:**
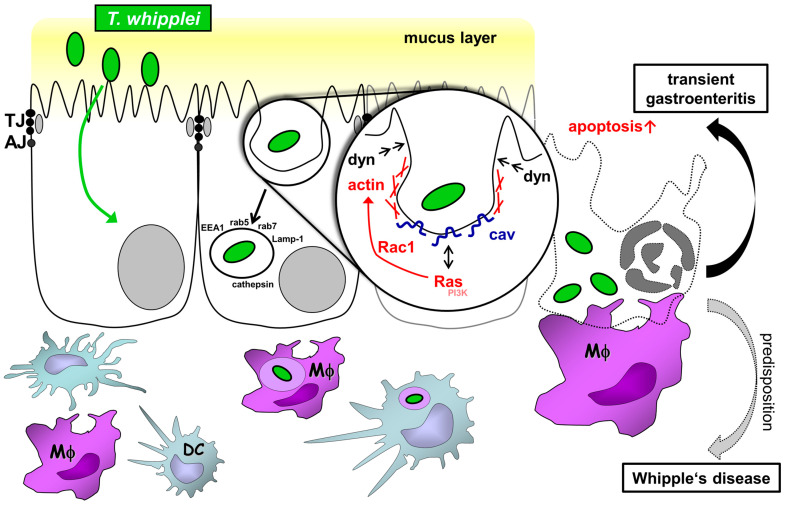
Scheme of the proposed mechanism for *T. whipplei* uptake and *T.-whipplei*-induced barrier defect. *Tropheryma whipplei* (*TW*) are Gram-positive rod-like bacteria that cause Whipple’s disease. The current assumption is that *TW* enters the host via the small intestinal mucosa, where it is found in mucosal macrophages. Mucosal inflammation results in mucosal flattening and malabsorption of nutrients. Disease progression can involve multiple organs, including joints (arthritis), the heart (endocarditis), or central nervous system (cerebral Whipple). Interestingly, *TW* is also a causative pathogen of transient gastroenteritis, as it was found in 15% of children with gastroenteritis. Thus, it not only accounts for severe pathologies but is also a prevalent pathogen. Nevertheless, the mechanism *TW* uses to invade and damage the small intestinal mucosa is incompletely understood. In the present work, we used model systems for Whipple’s disease to reveal the initial uptake mechanism of *TW* by intestinal epithelial cells (IECs). Specifically, IECs were shown to endocytose *TW* by a mechanism involving actin polymerization, the large GTPase dynamin, caveolin, and the small GTPases Ras/Rac1. Although caveolin was involved, it functioned as a signaling molecule inducing endocytosis by large vesicular structures rather than as the structural component of caveolae. Furthermore, heavily *TW*-invaded IECs underwent apoptosis, thereby causing an epithelial barrier defect (as a putative mechanism of acute gastroenteritis due to apoptosis-induced leak flux diarrhea). Consecutively, apoptotic IECs were shown to be subject to uptake by lamina propria macrophages and dendritic cells. Predisposed hosts with insufficient bacterial clearance mechanisms due to persistent exposure, specific immunological, and genetic alterations will develop Whipple’s disease.

## Data Availability

The data presented in this study are available on request from the corresponding author. Data from patients are not publicly available due to general data protection regulation.

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
