# Peer review of "Uptake of Tropheryma whipplei by Intestinal Epithelia"

_ijms, 2023, doi:10.3390/ijms24076197_

Round 1
Reviewer 1 Report
The manuscript by Friebel et al address aims at elucidating the route of entry of Tropheryma whipplei. Using different ex vivo human samples and cell culture models of intestinal epithelia, the authors suggest that T. whipplei enter IEC by endocytosis/macropinocytosis. T whipplei uptake is accompanied by decreased TER which may results from apoptosis of IEC. Finally apoptotic T. whipplei-infected IEC may be phogocytosed by mucosal macrophages. Although potentially interesting and well written, the data presented here do not always support the conclusions of the authors and several caveats have to be addressed.
Comments:
1/ Line 22 and line 42: Authors state that T. whipplei can cause two different pathologies. This statement is somehow oversimplified and not true. T. whipplei is responsible for a wide range of clinical entities: acute, eg gastroenteritis, chronic in selected individuals (localized, eg endocarditis or systemic, aka Classic Whipple’s disease). That should be acknowledged and corrected.
2/ When infecting biopsies in Ussing chambers, 10^8 bacteria were used, while when infecting cell monolayer, MOI of 100 was used. Why was such a high dose of bacteria used? Also, the calculation of the MOI for cell monolayer is hypothetical. Indeed, cells are cultured for 2 weeks in order to i) proliferate and ii) differentiate. This may explain the high variability of the results presented.
3/ In figure 2A and line 110: inhibition of actin polymerization results in a 50% decrease of bacterial uptake. This suggest that 50% of bacteria enter the cell independently of actin polymerization, how would the authors explain this point and what hypothesis could be formulated? Adding controls in this experiment may be important to rule out the hypothesis of incomplete actin polymerization inhibition.
4/ related to 3: authors measure epithelial uptake using flow cytometry (line 157 and 457). This should be detailed in the methods section as quantification using flow cytometry in that particular case of cell monolayer culture is not intuitive.
5/ line 122-124. Authors should show the subcellular fractionation experiment showing that in IEC T. whipplei is found in cav1- Cav2- rab5- rab-7 and cathepsin- positive fractions. Indeed, and as acknowledged by the authors in fig 2E, it seems that only few bacteria co-localized with caveolin-1.
6/ The western blot used to quantify caveolin1 and 2 expression (fig 2G) upon T. whiplei infection should be shown.
7/ line 152-153 and figure S2: the fact that only viable bacteria are readily phagocytosed suggest that bacterial uptake requires an active bacterial mechanism. However, heat-killed bacteria are obtained by heating at 100°C for 90 min. This has to be controlled. Indeed, such treatment may affect the particulate nature of the bacteria but also destroy the epitopes recognized by antibodies used to quantify bacterial uptake.
8/ It is unclear why colocalization with CathepsinD and Lamp 1 presented in fig 3B and 3C would confirm the endocytic uptake of T. whipplei.
9/ the exploration of the role of the mucus with NAC is interesting. These data suggest that as for numerous enteropathogens, the mucus favors bacterial adherence. However, bacteria must then express mucinase or “swim” in the mucus to reach the epithelial monolayers. In addition, it remains unclear for the reviewer if the mucus is required for bacterial uptake since polarized cell monolayer that do not produce mucus (Caco-2) are shown to be efficiently infected in the present manuscript. This should be discussed.
10/ The induction of IEC apoptosis by T. whipplei is also interesting and may explain the loss of resistance of the monolayers. However, it is unclear how apoptosis (and proliferation) were quantified. Authors also say that “IECs with a high load of TW were particularly prone to undergoing apoptosis-associated cell death” (line 219) and refers to Fig 5E. On which criteria can the authors say that this cell undergoes apoptosis? In addition, the link between apoptosis and loss of TER seems intuitively probable, however, the use of Z-VAD FMK only has a modest effect on the TER (fig 5C), although the p value appears very highly significant. If apoptosis cannot fully take in account the loss of TER, which mechanism could cause TER decrease?
11/ phagocytosis of infected apoptotic cells would be an elegant way used by T. whipplei to gain access to mucosal macrophages. However, this hypothesis only relies on the picture presented in Fig5F and fig S5. Other experiments should be conducted, especially regarding the coculture system used. Indeed, based on this protocol, the polarized IEC monolayer should be upside down, that is with the apical side towards the bottom. Does that mean that bacteria infect the IEC through the basolateral side? In addition, line 469-470, it is written “Filter inserts were then transferred to a 24-well plate (now with the upper filter side facing up) to seed macrophages (1.5 x 105 cells per filter inserts). On day 14, TW was added…”. It means that the coculture between IEC and macrophages last 13 days without TW, suggesting that macrophage may catch up some GFP-actin from Caco-2 cells, which may explain the staining pattern observed in fig 5F and S5, where it looks like macrophage are full of GFP-actin. This experiment definitely needs to be optimized to really show what the authors claim.
12/ line 362-364: authors raised an important point. Does T. whipplei promotes its own uptake? If viability is required that would suggest an active, de novo synthetized mechanism, such as the assembly of secretion systems. Authors mention that in the genome, 2 genes may be involved in the typeII/type IV secretion pathways. However, type II or type IV secretion systems are highly conserved and complex systems involving multiple genes (more than 20 for type II). Hence, although potentially interesting the presence of such systems in T. whipplei is unlikely, this should be acknowledged by the authors.
Author Response
Comment 1:
Line 22 and line 42: Authors state that T. whipplei can cause two different pathologies. This statement is somehow oversimplified and not true. T. whipplei is responsible for a wide range of clinical entities: acute, eg gastroenteritis, chronic in selected individuals (localized, eg endocarditis or systemic, aka Classic Whipple’s disease). That should be acknowledged and corrected.
Reply:
We thank the reviewer for this suggestion and have revised this part for clarity.
Comment 2:
When infecting biopsies in Ussing chambers, 10^8 bacteria were used, while when infecting cell monolayer, MOI of 100 was used. Why was such a high dose of bacteria used? Also, the calculation of the MOI for cell monolayer is hypothetical. Indeed, cells are cultured for 2 weeks in order to i) proliferate and ii) differentiate. This may explain the high variability of the results presented.
Reply:
We understand the reviewer’s concern about the methodology. In Ussing chamber experiments, TW was used at a final concentration of 107/ml (we have specified and corrected it from 108). Thank you for catching this error, which we might have caused confusion and which we now have corrected in the manuscript.
A MOI of 50:1 was established and used in several previous publication of groups very experienced in the cultivation of T. whipplei (https://doi.org/10.1128/IAI.70.3.1501-1506.2002 and https://doi.org/10.1038/cddis.2010.11) and the MOI of ≈ 100 that was used by us refers to the number of cells directly before incubating the monolayer with TW. The number of IECs was counted and calculated for each cell type (microscopy). We have added this information to the methods section of the manuscript. Nevertheless, the authors are aware of the inherent methodological limitations which we have discussed in the revised version of the manuscript.
Comment 3:
In figure 2A and line 110: inhibition of actin polymerization results in a 50% decrease of bacterial uptake. This suggest that 50% of bacteria enter the cell independently of actin polymerization, how would the authors explain this point and what hypothesis could be formulated? Adding controls in this experiment may be important to rule out the hypothesis of incomplete actin polymerization inhibition.
Reply:
The reviewer raises an interesting question. We agree that additional analyses would provide useful and important data, but we believe that the recommended analyses are outside the scope of this study. We propose that TW enters epithelia beyond an actin-dependent pathway, via a dynamin-, caveolin- and Ras-Rac1-dependent endocytosis mechanism. Furthermore, we suggested that TW causes IEC apoptosis primarily in IECs invaded by multiple TW bacteria. Therefore, apoptosis of IECs might promote transepithelial passage of TW independently of an endocytic pathway. We have now discussed this important aspect.
The authors assume – similarly to the reviewer - a higher level of complexity regarding the underlying pathophysiology. In this regard, it is likely that endocytic- and apoptotic pathways affect each other. Unfortunately, this cannot be sufficiently dissected by the static models used in our study. Since transmission of TW via the fecal-oral route is likely and the mechanisms TW takes advantage to pass the intestinal epithelial barrier were unknown, we used static models (with inherent limitations) to study selected aspects of initial steps of TW infection.
Comment 4:
Related to 3: authors measure epithelial uptake using flow cytometry (line 157 and 457). This should be detailed in the methods section as quantification using flow cytometry in that particular case of cell monolayer culture is not intuitive.
Reply:
We agree. In order with your suggestion, we have added a more detailed description to the manuscript with a specific focus on methodological aspects of the flow cytometric experiment. IEC cell suspensions were analysed by flow cytometry with rabbit-anti-TW (provided by Didier Raoult, Institut Hospitalo-universitaire Méditerranée Infection, Marseille, France; Aix Marseille Université, Institut de Recherche pour le Développement, Assistance Publique-Hôpitaux de Marseille, Microbes Evolution Phylogénie et Infections, Marseille, France), and visualized using the appropriate secondary donkey-anti-rabbit antibody linked to Alexa-488 or Alexa-647 (Life Technologies, 1:500). Flow cytometry was carried out using either a FACSCalibur or a FACSCanto II (both BD Biosciences) and analyzed with FlowJo software (FlowJo, Ashland, OR).
In addition, we prepared supplemental figures (S1, S2, S5, S8, and S10) that demonstrate the quantification of TW infected Caco-2 by flow cytometry.
Comment 5:
Line 122-124. Authors should show the subcellular fractionation experiment showing that in IEC T. whipplei is found in cav1-, cav2-, rab5-, rab7 and cathepsin- positive fractions. Indeed, and as acknowledged by the authors in fig 2E, it seems that only few bacteria co-localized with caveolin-1.
Reply:
We agree and now included the subcellular fractionation experiments (Fig. S3).
Comment 6:
The western blot used to quantify caveolin1 and -2 expression (fig 2G) upon T. whiplei infection should be shown.
Reply:
We agree and now added this information (Fig. S6).
Comment 7:
Line 152-153 and figure S2: the fact that only viable bacteria are readily phagocytosed suggest that bacterial uptake requires an active bacterial mechanism. However, heat-killed bacteria are obtained by heating at 100°C for 90 min. This has to be controlled. Indeed, such treatment may affect the particulate nature of the bacteria but also destroy the epitopes recognized by antibodies used to quantify bacterial uptake.
Reply:
We understand the reviewer’s concern on potential pitfalls of experiments on heat-inactivated/-killed bacteria. However, two recent studies also suggested that viable bacteria (heat killed bacteria were obtained by 80/100°C for 1h in these studies) are necessary for endocytic uptake of TW into macrophages as well as TW-induced apoptosis of macrophages (https://doi.org/10.1128/IAI.70.3.1501-1506.2002 and https://doi.org/10.1038/cddis.2010.11). Furthermore, it has been shown that immunohistochemistry is highly specific and sensitive and is applicable as a diagnostic method on intestinal tissue specimens to detect T. whipplei during active infection or in retrospective studies (https://doi.org/10.1016/S0046-8177(03)00126-6.) The TW-specific antibody applied in this study, is also used for routine clinical diagnostics in paraffin embedded specimens. We have added this information to the manuscript. For immunohistochemistry of paraffin embedded specimens, several steps at high temperature are applied that do not influence the binding of the polyclonal rabbit anti-TW antibodies. In detail: For the embedding in paraffin specimens are kept for several hours in pure melted paraffin at 70°C. For antigen retrieval, the sections are cooked in a high-pressure cooker even at 110°C and this procedure does not impede the binding of the antibody. Thus, one can assume that the process of heat-killing of TW does not destroy the epitopes recognized by the antibody used.
Comment 8:
It is unclear why colocalization with Cathepsin D and Lamp 1 presented in fig 3B and 3C would confirm the endocytic uptake of T. whipplei.
Reply:
The reviewer is correct, that this conclusion is drawn not only on the basis of our experiments but by integrating findings from previously published experiments into this disease model. Previous studies have elucidated that within macrophages, TW creates a niche by manipulating steps of phagosome–lysosome maturation, thereby facilitating its replication in an acidic environment, which is critical for its prolonged persistence in host cells during WD. We suspected a similar mechanism in IECs, since several proteins derived from the endolysosomal pathway were enriched in the TW-containing compartments. The presence of TW within Cathepsin- and LAMP1-positive compartments within IECs (of human biopsies) were not shown before. Fig. 3B and Fig. 3C reveal colocalization of TW (i) within IECs and (ii) LP macrophages with Cathepsin D and LAMP1. within line with your suggestion, we have moreover clarified the according paragraph of the manuscript.
Comment 9:
The exploration of the role of the mucus with NAC is interesting. These data suggest that as for numerous enteropathogens, the mucus favors bacterial adherence. However, bacteria must then express mucinase or “swim” in the mucus to reach the epithelial monolayers. In addition, it remains unclear for the reviewer if the mucus is required for bacterial uptake since polarized cell monolayer that do not produce mucus (Caco-2) are shown to be efficiently infected in the present manuscript. This should be discussed.
Reply:
The reviewer raises an interesting question. Mucolytic enzymes have not been described thus far for TW. However, two aspects need to be discussed in this context. Compared to monolayer experiments, where TW is exposed statically for 24 hours, during Ussing chamber experiments, permanent flow (generated by recirculation of apical fluid) might decrease adherence of TW after NAC pretreatment, thus protecting from TER reduction as observed in Fig. 3F. Pretreatment with NAC removed the apical mucus layer (Fig. 3E). TW was added after this step. Hence, TW had access to the epithelial layer (both, in the setting of mucus reduction [HT-29/B6 + NAC], or absence of mucus [T84, Caco-2, IPEC-J2] but also with the existing mucus layer [HT-29/B6 without NAC]), resulting in TER-reduction due to induction of IEC-apoptosis.
Furthermore, TW-specific staining (Fig. 3C) was performed after an initial washing step (already after the infection period of 24 h). Our data suggest that TW was flushed away from the apical monolayer surface (after NAC-pretreatment). But this, presumably, did not limit the access of TW to the epithelial layer during the prior infection period.
We have adjusted the text to clarify this point.
Comment 10:
The induction of IEC apoptosis by T. whipplei is also interesting and may explain the loss of resistance of the monolayers. However, it is unclear how apoptosis (and proliferation) were quantified. Authors also say that “IECs with a high load of TW were particularly prone to undergoing apoptosis-associated cell death” (line 219) and refers to Fig 5E. On which criteria can the authors say that this cell undergoes apoptosis? In addition, the link between apoptosis and loss of TER seems intuitively probable, however, the use of Z-VAD FMK only has a modest effect on the TER (fig 5C), although the p value appears very highly significant. If apoptosis cannot fully take in account the loss of TER, which mechanism could cause TER decrease?
Reply:
We thank the reviewer in supporting our intention. Previous studies have shown that TW also induces apoptosis in macrophages via caspase-3.
Apoptosis (caspase-3-positive) and proliferation index (Ki-67-positive) were expressed as the percentage of cleaved-caspase3-positive cells and Ki-67-positive cells, relative to the total number of epithelial cells per filter. Going in line with your suggestion, we have added this information to the legend of Fig. 5.
Detached cells were obtained from the apical supernatant of a T84 epithelial monolayer after incubation with TW. Criteria for apoptosis: rounded, single cell, cell shrinkage and nuclear fragments.
Fig. 5A reveals that Z-VAD rescued IECs from TW-induced apoptosis. Hence, one would expect a rescue of the TW-induced barrier defect (measured by TER, Fig. 5C). This, in the opinion of the authors, highlights that the barrier defect (measured by TER) is only partially explained by epithelial apoptosis (e.g. also by effects on tight junction protein expression) since it only partially affected by the pan-caspase inhibitor Z-VAD.
Comment 11:
Phagocytosis of infected apoptotic cells would be an elegant way used by T. whipplei to gain access to mucosal macrophages. However, this hypothesis only relies on the picture presented in Fig5F and fig S5. Other experiments should be conducted, especially regarding the coculture system used. Indeed, based on this protocol, the polarized IEC monolayer should be upside down, that is with the apical side towards the bottom. Does that mean that bacteria infect the IEC through the basolateral side? In addition, line 469-470, it is written “Filter inserts were then transferred to a 24-well plate (now with the upper filter side facing up) to seed macrophages (1.5 x 105 cells per filter inserts). On day 14, TW was added…”. It means that the coculture between IEC and macrophages last 13 days without TW, suggesting that macrophage may catch up some GFP-actin from Caco-2 cells, which may explain the staining pattern observed in fig 5F and S5, where it looks like macrophage are full of GFP-actin. This experiment definitely needs to be optimized to really show what the authors claim.
Reply:
Thank you very much for pointing this out. Macrophages and TW were added at the same time at Caco-2 culture day 14. We revised the manuscript to be clearer and pointed out that TW infection occurred via the basolateral side (Fig. S10).
We agree that additional analyses would provide useful and important data, but we believe that the recommended analyses are outside the scope of this study. Our intention was to describe selected aspects of TW infection that have been already described for antigen presenting cells (endocytosis and apoptosis), but not for IECs by using disease models. We included a separate figure that, in the opinion of the authors, underlines the importance of apoptosis of IECs for the uptake of TW (new Fig. S10). Given the small pore size of the filter membrane, macrophages (within the basal compartment) won’t have access to apoptotic, TW-loaded IECs. Furthermore, to ensure a direct contact of macrophages with the epithelial layer is not trivial.
However, we were able to show that: a) TW reaches antigen presenting cells after passage through an epithelial layer (Ussing chamber ex vivo model, TW is present in LP APCs, Fig. 1C-F and Fig. 5F and Fig. S12) b) apoptotic IECs are phagocytosed by macrophages in our co-culture model.
Several statements that we made in this regard were more ambiguous than intended. Thus we clarified those passages of the text and have stressed the limitations of the model.
Comment 12:
Line 362-364: authors raised an important point. Does T. whipplei promotes its own uptake? If viability is required that would suggest an active, de novo synthetized mechanism, such as the assembly of secretion systems. Authors mention that in the genome, 2 genes may be involved in the type II/type IV secretion pathways. However, type II or type IV secretion systems are highly conserved and complex systems involving multiple genes (more than 20 for type II). Hence, although potentially interesting the presence of such systems in T. whipplei is unlikely, this should be acknowledged by the authors.
Reply:
Thank you for catching this. We added this information to the discussion section.
Reviewer 2 Report
In this manuscript Friebel and colleagues investigate how TW invades the small intestinal epithelium and is taken up by phagocytic cells. The authors find that TW enters the epithelium in an actin, dynamin, and caveolin, and Ras-Rac1 dependent endocytic process, which subsequently causes epithelial cell apoptosis and barrier leak. The article is well articulated and the presented data supports the conclusions.
Concerns:
Current practice is to present data as scatter plots with means and error bars to rapidly inform the reader of replicates and variance.
In some places ‘data not shown’ should be included particularly with making the conclusions that the process is caveolin dependent.
Figure 2G is not a Western blot, but a quantitation of a Western blot. A representative image of the Western blot should be shown for clarity.
The legend and labelling of figure 5D are incongruent. Is this TW in green or caspase-3?
Minor concerns:
This reviewer was unable to view the supplemental movies.
Line 50 – Entering the small intestine in what way? The enterocytes or the lumen?
Line 61 – What is meant by autonomously? As in how the bacteria enter the enterocytes rather than being engulfed by immune cells?
Line 84 – In patients or in the issuing chamber experiment?
Line 114 – This needs rewording if it was not significant after multiple comparisons then it was not significant
Line 184 – A rationale for using E. coli K12 as a control should be given.
Author Response
Reviewer #2:
Comment 1:
Current practice is to present data as scatter plots with means and error bars to rapidly inform the reader of replicates and variance.
Reply:
Thank you for this suggestion. We have revised accordingly.
Comment 2:
In some places ‘data not shown’ should be included particularly with making the conclusions that the process is caveolin dependent.
Reply:
We agree and have revised this point accordingly.
Comment 3:
Figure 2G is not a Western blot, but a quantitation of a Western blot. A representative image of the Western blot should be shown for clarity.
Reply:
Thank you for catching this error, which we have now corrected in the manuscript.
Comment 4:
The legend and labelling of figure 5D are incongruent. Is this TW in green or caspase-3?
Reply:
Thank you for catching this. It has been corrected. Green = caspase-3.
Comment 5:
This reviewer was unable to view the supplemental movies.
Reply:
All movies were uploaded again.
Comment 6:
Line 50 – Entering the small intestine in what way? The enterocytes or the lumen?
Reply:
We have revised the manuscript: “….entering the small intestinal lamina propria (LP).”
Comment 7:
Line 61 – What is meant by autonomously? As in how the bacteria enter the enterocytes rather than being engulfed by immune cells?
Reply:
Our intention was to provide evidence that TW passes epithelial barrier without the help of immune cells (e.g. IEL). This was clarified in the new version of the manuscript.
Comment 8:
Line 84 – In patients or in the Ussing chamber experiment?
Reply:
Thank you very much for pointing this out. We have adjusted the text to clarify this point: “The number of stained rods was considerably lower in the samples exposed to TW ex vivo (Ussing chamber) compared to the mucosa of patients with established WD.”
Comment 9:
Line 114 – This needs rewording if it was not significant after multiple comparisons then it was not significant.
Reply:
We agree and have revised the manuscript accordingly.
Comment 10:
Line 184 – A rationale for using E. coli K12 as a control should be given.
Reply:
In the E. coli K12 strain, most virulence factor genes are absent that are present in intestinal and extraintestinal pathogenic E. coli strains. Genes for most adhesins as well as for toxins, O-antigens, glycocalyx proteins, invasins and other virulence factors are missing. E. coli K12 is unable to attach to the mucosa of the intestinal epithelium since no adhesins for this purpose are expressed, the production of O-antigens is impaired, and no capsule is formed. Symptoms of disease following oral ingestion of E. coli K12-derived strains have not occurred in volunteers or in laboratory animals (Statement of the ZKBS on the suitability of Escherichia coli K12-derived strains as part of biological safety measures according to § 8 para. 1 GenTSV). We have added this information to the manuscript.

Round 2
Reviewer 1 Report
Authors have addressed most of the concerns raised by the reviewer. However, the quantification of T. whipplei by flow cytometry needs additional controls. It is difficult to understand why T. whipplei does not constitute a clear distinct population on the dot plots (see fig S1, S2 and S5). In addition, the string seems very dim. Thus, in order to support the data, isotype controls need to be provided.
Also, in the fractionation experiment depicted in fig S3, authors claimed that T. whipplei is found inwithin fractions positive for Rab5, Rab7 and cathepsin D. However, in the fraction were T. whipplei is the most present (fraction 8) the Rab5 does not seem to be expressed (same for fraction 9). Given that T. whipplei replicates in vacuoles that coexpress Rab5 and Rab7, what does that mean in this setting ?Do the authors have an explanation for that?
Author Response
Comment 1:
The quantification of T. whipplei by flow cytometry needs additional controls. It is difficult to understand why T. whipplei does not constitute a clear distinct population on the dot plots (see fig S1, S2 and S5). In addition, the string seems very dim. Thus, in order to support the data, isotype controls need to be provided.
Reply:
We thank the reviewer in supporting our intention. We observed that the number of stained rods was considerably lower in the samples exposed to TW ex vivo (Ussing chamber) compared to the mucosa of patients with established WD. This was an expected finding, given the relatively short period of TW exposure in the Ussing (and monolayer) experiment compared to the length of exposure in an individual infected with TW, and the biology of TW itself. Furthermore, we observed (monolayer experiments) that IECs were infected infrequently (Fig. 1G-1J, Fig. 2B, 4C).
By using flow-cytometry was, in the opinion of the authors, the most reliable method with the best sensitivity and specificity. The monolayer experiments revealed a heterogeneous pattern of TW-infected cells after 24h. Only few cells engulf TW and the amount of TW per cells varies markedly. This heterogeneity, together with the infrequency, explains that no clear distinct TW infected population can be separated. We have now added this information/limitation to the method section. Interestingly, confocal microscopy (Fig. 1H) as well as TEM revealed that IECs with a high load of TW were particularly prone to undergoing apoptosis-associated cell death. And cells in an advanced state of apoptosis with a high load of TW might get lost during the procedure of preparation for FACS-Analysis.
In order with your suggestion, we now have included an additional control (Fig. S13). Caco-2 cells were incubated with vital TW and stained for flow cytometric analysis with A: rabbit serum and donkey- anti-rabbit-Alexa647 as isotype control or B: rabbit anti-T. whipplei followed by donkey-anti-rabbit-Alexa647.
Comment 2:
Also, in the fractionation experiment depicted in fig S3, authors claimed that T. whipplei is found in within fractions positive for Rab5, Rab7 and cathepsin D. However, in the fraction were T. whipplei is the most present (fraction 8) the Rab5 does not seem to be expressed (same for fraction 9). Given that T. whipplei replicates in vacuoles that coexpress Rab5 and Rab7, what does that mean in this setting? Do the authors have an explanation for that?
Reply:
Thank you for catching this. We now highlighted this information in the figure legend. Previous studies have shown that TW inhibits phago-lysosome biogenesis to create a suitable niche for its survival and replication in macrophages. Mottola et al. showed that in addition to Lamp-1, the TW phagosome was positive for Rab5 and Rab7 in mouse bone marrow macrophages. In our study, the compartments containing TW were recovered in fractions 3-10 of the sucrose gradient, with a focus in fraction 8. The fact that TW accumulates in fraction 8, which has a low expression of Rab5 does, in the opinion of the authors, not automatically imply that the endocytic process is independent of this GTPase. Endocytosis with phagosome maturation and phagosome transition is a rather dynamic process and it might differ in mouse bone marrow macrophages and human epithelial cells that are no professional cells for the uptake of bacteria. Subcellular fractionation was done after an incubation period of 24h. Therefore, it is likely that at this timepoint the maturation process is already beyond the initial steps that require Rab5 or that the possibilities of TW to block the transition from Rab5 to Rab7 are more limited in human epithelial cells. This is in accordance with the finding that TW co-localized with the late endosomal and lysosomal markers Lamp-1 and cathepsin in IECs (Fig. 3B and C).
